# Synthesis, Structure and Supramolecular Properties of a Novel C3 Cryptand with Pyridine Units in the Bridges

**DOI:** 10.3390/molecules25173789

**Published:** 2020-08-20

**Authors:** Cosmin Vasile Crişan, Albert Soran, Attila Bende, Niculina Daniela Hӑdade, Anamaria Terec, Ion Grosu

**Affiliations:** 1Department of Chemistry and SOOMCC, Faculty of Chemistry and Chemical Engineering, Babes-Bolyai University, Cluj-Napoca, 11 Arany Janos str., 400028 Cluj-Napoca, Romania; kossminn@yahoo.com (C.V.C.); albert_soran@yahoo.com (A.S.); bogdan.niculina@gmail.com (N.D.H.); asuciu@chem.ubbcluj.ro (A.T.); 2National Institute for Research and Development of Isotopic and Molecular Technologies, 67-103 Donath str., RO-400293 Cluj-Napoca, Romania; attila.bende@itim-cj.ro

**Keywords:** cryptand, 1,3,5-triazine, single crystal X-ray structure, theoretical calculations, host–guest entities

## Abstract

The high-yield synthesis and the structural investigation of a new cryptand with C3 symmetry, exhibiting 2,4,6-triphenyl-1,3,5-triazine central units and pyridine-based bridges, are reported. The structure of the compound was investigated by single crystal X-ray diffractometry, NMR (nuclear magnetic resonance), HRMS (high resolution mass spectrometry) measurements, and theoretical calculations. The study of supramolecular behavior in solid state revealed the association of cryptand molecules by C-H---π and π---π contacts. Moreover, theoretical calculations indicated the high binding affinity of the cryptand for various organic molecules as guests.

## 1. Introduction

Cryptands exhibiting C_3_ symmetry have been exciting synthetic targets, due to their high ability to form stable “host–guest” supramolecular assemblies (i.e., as result of the macrobicyclic effect [1]) either with ions (cations and anions) [2,3,4] or neutral organic molecules [5,6,7,8,9]. Many aromatic tripodand units were employed to access this type of cryptand. Thus, bicyclocyclophanes exhibiting 1,3,5-trisubstituted benzene [10,11,12,13,14], 2,4,6-trisubstituted-1,3,5-triazine [15,16,17], tertiary amines [18,19], or phosphines [20,21], as well as cyclotribenzylene [22,23,24] as central units, connected by various bridges, such as oligoethyleneoxide [25,26], di-yne [17,27,28], pyridine [29,30,31], or triazole [32,33,34,35], in agreement with the employed macrocyclization procedure, have been reported and showed good binding abilities for various guests.

1,3,5-Triazine is a largely investigated heterocycle, on one side due to the versatile synthesis ofits many derivatives (e.g.,acid-catalyzedtrimerization of nitriles, O-substitution of cyanuric acid, or N-substitution of isocyanuric acid) and on the other side due to the exciting properties and applications of 1,3,5-triazine’s derivatives (e.g., melamine resins in the furniture industry) [36,37,38].

In previous works, we investigated several C_3_ cryptands (Scheme 1, **1**–**3**) exhibiting 1,3,5-triazine or 2,4,6-triphenyl-1,3,5-triazine central aromatic units. Cryptand **1** was obtained by an acetylenic coupling reaction and, despite our expectation to have a shape-persistent compound with a large internal cavity, the single crystal X-ray diffraction molecular structure of cryptand **1** showed the completely collapsed cavity behavior of the cryptand [17]. Compounds **2**(**a**–**c**) exhibit a high affinity for alkali (Na^+^, K^+^) and ammonium cations (e.g., dication of 1,5-naphthalenediamine). This complexation behavior is due to the ethyleneoxide groups in the bridges and, in the investigated series, the affinity for K^+^increased with the number of ethyleneoxide units [2]. High-affinity, charge–transfer complexes of compound **3**, based on the electron deficient 1,3,5-triazine units (with electron acceptor ability) with electron rich aromatic guests (pyrene, anthracene, 1,5-dinaphthol) were reported, where π–π stacking contacts ensured the formation of the host–guest complexes. No complexations via H-bonds involving the pyridine rings of the bridges were observed, confirming the expected low basicity of pyridine groups due to the powerful electron-withdrawing CN substituents attached to the heterocycles in the chains [7]. In this context, we considered it of interest to investigate the structure and properties of cryptand **4**, exhibiting more basic pyridine rings, that should have better proton and halogen binding affinity, along with slightly more flexible arms than those in the case of the previously investigated cryptand **3**.

## 2. Results and Discussion

### 2.1. Synthesis and Structure of Cryptand 4

Cryptand **4** was obtained in good yields (64%) by a nucleophilic substitution reaction using 1,3,5-triazine-2,4,6-triphenol **5 [39,40,41]** as a nucleophile and 2,6-dibromomethyl-pyridine as a substrate (Scheme 2). The procedure was adapted from the literature using our previous experience with similar reactions [2,7,8,42,43].

The structural investigation of cryptand **4** was carried out using NMR, HRMS (high resolution mass spectrometry), and single crystal X-ray diffractometry. Theoretical calculations were performed to estimate the contribution of different crystal fragments to the crystal cohesive energy. In order to fulfill this target, fragments were extracted from the lattice obtained by single crystal X-ray diffraction, and the calculated energy of the assembly of the extracted molecules was compared to the energy of the system exhibiting the same molecules in isolated behavior. The second goal of theoretical calculations was to estimate the binding ability of the cryptand towards some representative organic molecules as guests. The NMR spectra of **4** were very simple and confirmed the C_3_ symmetrical structure. Thus, the ^1^H NMR spectrum of 4 (Figure 1) revealed one singlet (δ = 5.35 ppm (parts per million)) for all CH_2_ groups, while in the aromatic region of the spectrum there were two doublets (δ = 6.81 and 7.90 ppm) for the protons of the 1,4-phenylene groups and the classic pattern for the symmetrically 2,6-disubstituted pyridine unit (doublet (δ = 7.53 ppm) and triplet (δ = 7.92 ppm)). The ^13^C NMR spectrum (APT (attached proton test), Figure 1) exhibited only nine signals, five on one side for CH_2_ (δ = 69.32 ppm) and quaternary aromatic carbon atoms (δ = 127.89, 158.04, 161.29, 169.56 ppm) and four signals on the other side belonging to the tertiary aromatic carbon atoms (δ = 114.58, 121.78, 129.52, 137.75 ppm).

Single crystals of **4** were obtained at room temperature by vapor diffusion between a dichloromethane solution of **4** and ethanol.

The X-ray diffraction on a single crystal of compound **4** revealed an asymmetric unit with one molecule (Figure 2). In this structure, the two 1,3,5-triazine units were slightly rotated and parallel to each other with an angle between the planes defined by these rings of 4.5 (1)°, an average dihedral angle C−C_g_−C_g_’−C or N−C_g_−C_g_’−N of 36.3(1)° and a C_g_−C_g_’ distance of 3.35 Å, which suggests high π···π interactions (where C_g_ and C_g_’ are the centroids of the two triazine rings) (Figure 3a). Due to the rotation and orientation of the *p*-phenylene units relative to the 1,3,5-triazine rings, a propeller-like arrangement can be identified within the molecule, with the unit cell containing two molecules, one with left- and the other one with a right-handed propeller arrangement (Figure 3b). The left/right-handed propeller arrangement was, however, consistent only for the C1−N1−C2−N2−C3−N3 triazine unit but not for the C23−N4−C24−N5−C25−N6 unit.

In the crystal of **4**, several weak C−H···O and C−H···N hydrogen interactions were found (see Figure 4, Figure 5 and Figure 6). [44] The numerical data for these interactions are given in Table 1.

Furthermore, several C−H···π and π···π intermolecular interactions were found in the crystal of **4** (see Figure 7 and Figure 8 and numerical data in Table 1) [44,45,46].

The associations of cryptand molecules, either by π---π contacts with an assembly (named a “stacked” pair) or by C-H---π interactions with the so called “side” pair, are shown in Figure 9. For the stacked pair, the 2,4,6-triphenyl-1,3,5-triazine central units partially overlapped, so the two triazines formed stacking contacts with the two benzene rings. The intermolecular binding energy between the two cryptands in this stacking supramolecular structure was −28.77 kcal/mol. In the case of the side-pair superstructure, two benzene rings from the phenyl fragments of two different cryptands form T-shape contacts with the aromatic rings of the pyridine fragments of the bridge. In these two T-shape contacts, the distance between the H atoms of the benzene units and the centroids of the pyridine rings was 2.99 Å, while the whole intermolecular binding energy, where these T-shape structures were included, was −18.60 kcal/mol. 

### 2.2. Complexation Abilities of Cryptand **4**

In the next step, the ability of the cryptand to bind guest molecules into the cavity, defined by the two 2,4,6-triphenylene-1,3,5-triazine central units was investigated. Accordingly, four guest molecules (G1, G2, G3, and G4; Figure 10), chosen for their potential ability to interact by hydrogen or halogen bonding with pyridine units in the bridges of the cryptand, as well as to form charge–transfer or π–π-associated complexes with the central units, were investigated.

It is known from our previous experience [7] that the first step inthe inclusion kinetics, when the cavity is being opened, is quite slow and cumbersome, since, at the beginning of the intercalation, the excess energy obtained from the interaction cannot fully compensate for the increased energy required to deform the cryptand. To overcome this problem, the initial geometries of the host–guest complexes were set with the guest molecule fully inserted between the central units of the cryptand. In this sense, two energy components are particularly important. One of them is the intermolecular binding energy between the cryptand and the guest molecules, while the second is the deformation energy of the cryptand. The host–guest supramolecular structure of the cryptand–G1 complex is shown in Figure 11a,b. Surprisingly, the interplanar distances between the guest system and the two 2,4,6-triphenyl-1,3,5-triazine central units of the cryptand were not similar. They were 3.38 Å and 3.77 Å, respectively, while, due to the intercalation process, the cryptand’s two central units departed from 3.40 Å to 7.15 Å. The binding energy between the cryptand and the G1 guest molecule was -56.76 kcal/mol, while the deformation energy of the cryptand from its equilibrium geometry was +24.21 kcal/mol. Accordingly, the energy balance of this complexation was −32.55 kcal/mol. It is noteworthy that, in the host–guest complex, the pyridine bridge fragments were barely involved. The second G2 guest molecule had almost the same behavior as the one found for the G1 case. The host–guest configuration of the cryptand–G2complex is depicted in Figure 11c,d. The binding energy between the cryptand and the G2 guest molecule was −56.60 kcal/mol, while the deformation energy of the cryptand from its equilibrium geometry conformation was +21.44 kcal/mol. Comparing cases G1 and G2, the role of triazine in the guest molecule can be clearly demonstrated. Since the peripheral aromatic rings were not twisted so much with respect to the central triazine ring, the deformation of the cryptand became slightly smaller (with 2.77 kcal/mol) and the interplanar distance between the cryptand 1,3,5-triazine units shorter (6.68 Å). The interplanar distances between the guest molecule and the cryptand central units, taken at the central rings, were 3.27 Å and 3.39 Å, respectively. 

In order to check whether the functional group at the end of the peripheral arms of the guest complex could enhance the effect of the intercalation, in the case of the G3 guest system, the OH (targeted to be involved in hydrogen bond with the pyridine rings of the bridges) ending groups were changed to iodine atoms (able to form N---I halogen bonds). The host–guest structure of the cryptand–G3 complex is shown in Figure 11e,f. The binding energy between the cryptand and the G3 guest molecule was −53.13 kcal/mol, while the deformation energy of the cryptand from its equilibrium geometry was +21.21 kcal/mol. As one can observe, there is almost no energetic influence of the ending group on the stability of the intercalated guest molecules into the cryptand.

The interplanar distances between the guest molecule and the two cryptand central units, taken at the central rings, were 3.31 Å and 3.35 Å, respectively, while the interplanar distance between the two caps of the cryptand was 6.66 Å. It is known that fluorinated iodobenzene rings show stronger stacking interaction and stronger halogen bonds [47]. Accordingly, the 1,4-diiodotetrafluorobenzene (G4) was considered as the next guest structure and its intercalation properties were investigated. The host–guestconfiguration of the cryptand–G4 complex is depicted in Figure 11g,h. As it can be observed, the cryptand and the guest system showed a parallel-displaced arrangement and no halogen bonds were formed. The binding energy between the cryptand and the G4 guest molecule was −32.51 kcal/mol, while the deformation energy of the cryptand from its equilibrium was +19.72 kcal/mol. This binding can be considered as relatively strong, since the guest molecule contains only one aromatic ring and not four of them, as was the case for the previous three guests. The interplanar distances between the guest molecule and the cryptand 1,3,5-triazine units were 3.23 Å and 3.34 Å, respectively, while the interplanar distance between the two 1,3,5-triazine units of the cryptand was 6.57 Å.

The energetic study of the fitting of the guest molecules inside the cryptand cavity has shown that all four molecular structures chosen as possible guest molecules for the cryptand cavity are suitable. The binding energies rank between −32 and −56 kcal/mol, which are large enough to overcome the relatively high deformation potential of the cryptand. The complexation of **4** with different organic molecules is mainly due to the stacking (aromatic–aromatic) contacts, and no-binding via the hydrogen or halogen bond contacts was revealed.

Despite the lack of hydrogen and halogen bonds in the theoretical previsions concerning the binding abilities of cryptand **4** versus the investigated host–guest systems (G1–G4), the theoretical calculations were encouraging, and we carried out NMR and ESI (electrospray ionization)-MS experiments in order to reveal the formation of the supramolecular associations of cryptand **4** with G1, G2, and G5 (the equivalent of G3 but exhibiting Br instead of I atoms). Unfortunately, these experiments showed no clear complexation results, and we concluded that the kinetic barriers for the formation of the host–guest complexes are too high and other methods and experiments have to be envisaged for the future complexation investigations of cryptand **4**. 

## 3. Materials and Methods 

### 3.1. General Data

The ^1^H NMR (600 MHz) and ^13^C NMR (150 MHz) spectra were recorded in DMSO (dimethylsulfoxide)-d_6_ at *rt*, using the solvent line as a reference. The atmospheric pressure chemical ionization (APCI, positive ions mode) was recorded on an LTQ ORBITRAP XL spectrometer (Thermo Scientific) using external mass calibration. The melting point was measured with a routine apparatus. The triphenol triazine **5** and guests G1 and G5 were obtained using procedures described in the literature [39,40,41], while the other reagents were commercially available and used without further purifications. Thin layer chromatography (TLC) was conducted on silica gel 60 F254 TLC plates (Merck). The solvents were dried and distilled under argon using standard procedures.

Single crystals of **4** were obtained at room temperature by vapor diffusion between a dichloromethane solution of **4** and ethanol. The crystals were mounted on MiTeGen microMountscryoloops and data were collected on a Bruker D8 VENTURE diffractometer, using Mo-Kα radiation (λ = 0.71073 Å) from a IμS 3.0 micro focus source with multilayer optics at a low temperature (103 K). For structure solving and refinement, the Bruker APEX3 software package was used [48]. The structure was solved by dual methods (SHELXT-2014/5) [49] and refined by full matrix least-squares procedures based on *F^2^* with all measured reflections (SHELXL-2018/3) [50]. The structures were refined with anisotropic thermal parameters for non-H atoms. Hydrogen atoms were placed in fixed, idealized positions and refined with a riding model and a mutual isotropic thermal parameter. Several highly disordered and partially occupied dichloromethane molecules were found in the crystal structure, and their contribution to the electron density map was calculated and removed from the final model using PLATON-SQUEEZE [51,52]. The structure was deposited at the CCDC (Cambridge Crystallographic Data Centre) and the deposition number is 2019839.

Further details on the data collection and refinement methods can be found in Table 2. The drawings were created with the Diamond program [53]. The Appendix A for this paper can be obtained free of charge from the Cambridge Crystallographic Data Centre via https://www.ccdc.cam.ac.uk/structures/.

The equilibrium geometries and the intermolecular binding energies (ΔE in kcal/mol) for different host–guest assemblies between the cryptand **4** and G1–G4 have been obtained at density functionaltheory level, using the Head–Gordon’s density functional with minimal Fock exchange, called the ωB97X [54] exchange-correlation functional, including the Grimme’s empirical dispersion correction with Becke–Johnson damping (D3BJ) [55,56], as well as the def2-TZVP [57] basis set, together with the def2/J [58] auxiliary basis, as it is implemented in the Orca program package [59,60].

### 3.2. Procedure for the Synthesis of Cryptand **4**

A solution of 2,6-bis(bromomethyl)pyridine (78 mg, 0.294 mmol) and triphenol **5** (70 mg, 0.196 mmol) in tetrahydrofuran (5 mL) was added, dropwise and under stirring, to a mixture of potassium carbonate (270 mg, 1.96 mmol) and refluxing acetone (70 mL). The reflux of the solvent was continued overnight. After cooling to room temperature, water (50 mL) was added to the reaction mixture and the organic compounds were extracted with chloroform (3 × 25 mL). The crude product was washed with tetrahydrofuran (3 × 5 mL) to give pure cryptand **4** in 64% yields.

10,18,32,40,50,58-hexaoxa-2,4,24,26,45,63,66,71,76-nonaazatridecacyclo[25.17.2.^6,9^2.^19,22^2.^28,31^2.^41,44^2.^46,49^2.^59,62^1.^1,5^1.^12,16^1.^25,29^1.^34,38^1.^52,56^]octaheptaconta-1,3,5(45),6,8,12,14,16(71),19,21,23,25,27(63),28,30,34,36, 38(66),41,43,46,48,52,54,56(76),59,61,64,67,69,72,74,77-tricontaene **4**

White solid, mp = 211–212 °C (decomposition), ^1^H NMR (600 MHz, DMSO-d6, ppm) δ 7.93 (t, *J* = 7.8 Hz, 3H), 7.90 (d, *J* = 8.8 Hz, 12H), 7.53 (d, *J* = 7.8 Hz, 6H), 6.81 (d, *J* = 8.8 Hz, 12H), 5.35 (s, 12H); ^13^C NMR (150 MHz, DMSO-d6, ppm) δ 169.56, 161.29, 156.04, 137.75, 129.52, 127.89, 121.78, 114.58, 69.32.

HRMS(APCI; Atmospheric pressure chemical ionization) calculated for C_63_H_46_N_9_O_6_ [M+H]^+^: 1024.3566; found:1024.3550

## 4. Conclusions

The structural investigations and theoretical calculations revealed significant stacking abilities for cryptand **4**, which led to the formation in the crystal structure of two types of dimeric associations with a large contribution to the stability of the solid state structure. The theoretical calculations concerning the complexation of **4** with triphenols G1 and G2 and halogenated guests G3 and G4 showed the high stabilization of the complexes of cryptand **4** by aromatic–aromatic contacts, and no contributions by hydrogen or halogen bonds were observed. Despite the high stability of the investigated host–guest systems, there are huge energy barriers for the formation of these complexes, due to the deformation energy requested by the change of the shape of the cavity and its preparation for interaction with the guests. These high complexation barriers explain the failure of the NMR and ESI-MS investigations concerning the complexation abilities of cryptand **4**.

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
