# Peer review of "Synthesis, Structure and Supramolecular Properties of a Novel C3 Cryptand with Pyridine Units in the Bridges"

_molecules, 2020, doi:10.3390/molecules25173789_

Round 1
Reviewer 1 Report
The paper reports the synthesis of bicyclic cyclophane containing triazine and pyridine units, characterization, and binding study using molecular calculations. I like the paper because all data are fine. However, I need to mention some points to make the paper more high-quality.
In general, the compound such type is NOT classified as a cryptand. Cryptand means bicyclic crown ethers with nitrogen atoms at the bridgehead, and compound should be classified as a bicyclic cyclophane. The authors always calculate complexation energy instead of binding constants. If the estimated complexation energy between the host and guests are 30-50 kcal/mol, binding constants should be measured using some titration method.
I will recommend this manuscript be published after the following suggestions are satisfied; (i) measurement of the binding constant and (ii) elemental analysis of compound 4.
Reviewer 2 Report
The manuscript “Synthesis, structure and supramolecular properties of a novel C3 cryptand with pyridine units in the bridges" by Crişan, Soran, Bende, Hӑdade, Terec and Grosu presents the synthesis of a cavity containing cryptand from 2,4,6-triphenyl-1,3,5-triazine connected by pyridine bridges. They presented the crystal structure of this complex. Theoretical calculations show this compound to bind certain guest compounds. The results reveal a net stabilization of the host-guest complex with respect to the deformation of the cryptand. Unfortunately, no experimental results could back the theoretical calculations. This is a well-done work but there are clearly some deficiencies.
Major
- The synthetic protocol is well known, the difference between the already published compound 3, and the new compound 4 is very minor. This looks like unnecessary fragmentation of the work.
- The experimental evidence for host-guest is non-existence which makes this reviewer wonder, what is the purpose/reason to present the theoretical calculations?
- No supplemental information (SI) is provided. Not even the Checkcif file. The theoretical calculation coordinates are also missing (usually in the SI).
- References should be revised. For example, check reference 1………
- In the introduction: “In this context, we considered of interest to investigate the access and the properties of cryptand 4 exhibiting pyridine rings with better proton and halogen binding affinity than the previously investigated cryptand 3.” There is no halogen bonding in this work. This sentence is unnecessary and holds no context to the work.
Minor:
Some general language check. A few typos and awkward sentences here and there. E.g.
- Abstract: put a space between “measurementsand”
- Page 1: “due their” should be “due to their”
- Page 1: “bicylocyclophanes” should be “bicyclocyclophanes”
- And so on…….
The only true novelty here is the new crystal structure of compound 4. Is this enough to warrant publication to molecules? Considering that the synthetic protocol is exactly the same as reported in ref 8 (not 7) with compound 3, I think the authors should enhance the scope of this synthetic protocol, for example, include other pyridine derivatives, or get some guests that could be bound by this host. In the current form, this is very limited experimental work. Unnecessary fragmentation of work does not bring a broader picture and lacks a comparison aspect, making dissemination of knowledge across community more difficult.
Round 2
Reviewer 1 Report
This revised manuscript met the reviewer's request, so I recommend publishing it in Molecules.
Reviewer 2 Report
The authors did a good job with the review.
For the references, it seems when converting to Pdf the references are offset which is still the case with this version. I think it is a conversion issue and I urge the authors to check the proofs carefully when they are ready.
Though I would have liked the work more if more than one compound was synthesized, I now think it should be accepted.